# Surface Modification of Bamboo Charcoal by O_2_ Plasma Treatment and UV-Grafted Thermo-Sensitive AgNPs Hydrogel to Improve Antibacterial Properties in Biomedical Application

**DOI:** 10.3390/nano11102697

**Published:** 2021-10-13

**Authors:** Shih-Ju Liu, Shu-Chuan Liao

**Affiliations:** 1Design and Materials for Medical Equipment and Devices, Da-Yeh University Changhua, Changhua 515006, Taiwan; liuliu860501@gmail.com; 2Department of Biomedical Engineering, Da-Yeh University Changhua, Changhua 515006, Taiwan

**Keywords:** bamboo charcoal, O_2_ plasma treatment, thermo-sensitive AgNPs hydrogels, antibacterial

## Abstract

With the advancement of science and modern medical technology, more and more medical materials and implants are used in medical treatment and to improve human life. The safety of invasive medical materials and the prevention of infection are gradually being valued. Therefore, avoiding operation failure or wound infection and inflammation caused by surgical infection is one of the most important topics in current medical technology. Silver nanoparticles (AgNPs) have minor irritation and toxicity to cells and have a broad-spectrum antibacterial effect without causing bacterial resistance and other problems. They are also less toxic to the human body. Bamboo charcoal (BC) is a bioinert material with a porous structure, light characteristics, and low density, like bone quality. It can be used as a lightweight bone filling material. However, it does not have any antibacterial function. This study synthesized AgNPs under the ultraviolet (UV) photochemical method by reducing silver nitrate with sodium citrate. The formation and distribution of AgNPs were confirmed by UV-visible spectroscopy and X-ray diffraction measurement (XRD). The BC was treated by O_2_ plasma to increase the number of polar functional groups on the surface. Then, UV light-induced graft polymerization of N-isopropyl acrylamide (NIPAAm) and AgNPs were applied onto the BC to immobilize thermos-/antibacterial composite hydrogels on the BC surface. The structures and properties of thermos-/antibacterial composite hydrogel-modified BC surface were characterized by Scanning Electron Microscopy (SEM), Fourier Transform Infrared spectrum (FT-IR), and X-ray photoelectron spectroscopy (XPS). The results show that thermos-/antibacterial composite hydrogels were then successfully grafted onto BC. SEM observations showed that the thermos-/antibacterial composite hydrogels formed a membrane structure between the BC. The biocompatibility of the substrate was evaluated by Alamar Blue cell viability assay and antibacterial test in vitro.

## 1. Introduction

More and more biological resources are being used to prepare adsorbents, and some studies have shown that specific biological resources have a high potential for use as adsorbents [1,2,3,4]. Among them, bamboo charcoal is one of the most popular biological resources. With a porous structure, bamboo charcoal is good in adsorption. The main functional groups of bamboo charcoal are C=C, C-H, and a few -OH; it has excellent adsorption capacity for non-polar substances [3,4]. At the same time, bamboo charcoal also has several beneficial characteristics, including electromagnetic shield material, far infrared rays, non-toxic, bioinert material, and high electric conductivity [5,6,7,8]. It has been used in various fields such as environmental protection, biological materials, medicine, etc. However, its surface strength is not good and it is hydrophobic. It does not have any antibacterial function, thus limiting biomedical applications such as blood purification, bone repair or replacement, and neuroprotection [8,9,10,11]. In recent years, to pursue a safer and healthier environment while using antimicrobial agents, in addition to the safety of use, the protection of the environment must also be considered [12,13,14]. Nanoparticles (NPs) are of extraordinary interest due to their nanoscale size and large surface area, making them suitable candidates for various applications [13,14,15]. Additionally, they are useful for many applications. However, there are a few health hazards due to their uncontrollable use and discharge to the natural environment. 

In recent centuries, silver and its compounds have been widely used commercially. Silver ions have strong inhibitory and bactericidal effects, as well as broad-spectrum antibacterial activities [5,15]. Silver ions must be released into the pathogenic environment to sterilize effectively. When silver ions are excreted in negatively charged bacteria, the silver ions combine with the chemicals in the bacteria, causing the cells to die and achieve the purpose of inhibition naturally. When the cell death ruptures, the silver ions will be set free from the cells and continue to penetrate other bacteria [16,17,18,19,20,21]. As nanoparticles have unique physicochemical properties and a growth inhibitory capacity against microbes, researchers started to conduct experiments on the effectiveness of nanoparticles as antimicrobials [16,22,23,24,25]. Their antimicrobial efficacy is due to the large surface area to volume ratio and their chemical properties, which greatly benefit against drug resistance. The application of nanoparticles as antimicrobials is gaining significant interest in prevention and therapeutics in medical devices, the food industry, and textile fabrics [13,14,26,27,28]. Currently, there are physical, chemical, or biosynthesis methods for nanoparticle synthesis. Among them, chemical synthesis is the most used method to synthesize silver nanoparticles [29,30,31,32,33,34].

A plasma surface treatment system is a unique modification technology that is an advantageous and versatile technique in which plasma activation treatment and surface graft polymerization achieve uniformity and function [35,36,37,38]. Plasma treatment is used to alter the surface properties to increase the adhesion, wettability, and other surface characteristics of various materials [37,39,40,41]. Compared with most other treatments, a significant advantage of plasma surface modification, compared with most other treatments, is that it is free of harmful sub-products from the operation process and does not destroy the bulk structure of materials [37,42,43]. To enhance the different applications of materials, surface modification plays a critical role in various fields such as optical, electronics, automotive, aerospace, textile industry, packaging technology, 3D printing, pharmaceutical, and medical [42,43,44,45,46,47,48,49].

Hydrogel is a hydrophilic polymer with a three-dimensional network structure. After absorbing a large amount of water, it will expand and swell. Polymerized -N-isopropyl acrylamide (PNIAAm) is a well-known thermo-responsive polymer polymerized from the monomer NIPAAm [50,51,52]. PNIPAAm is a thermo-sensitive hydrogel with hydrophilic and hydrophobic groups which exhibits a lower critical solution temperature (LCST) of approximately 32 °C in an aqueous media. PNIPAAm-based hydrogel absorbs water and exist in swollen states below the LCST (<32 °C) [47,48,49,50,51,52,53]. However, when the temperature of the medium rises above the LCST (>32 °C), its volume will shrink suddenly and drastically [50,51,52,53,54,55,56]. This phase transition phenomenon is reversible [50,51,52,53,54,55,56,57]. According to the literature, it is found that the thermal response characteristics of PNIPAAm are related to the degree of grafting [55,58,59]. This transition has substantial importance and has allowed the use of PNIPAAm-based hydrogels in biomedical applications, such as in drug release, tissue engineering, cell culture, and electroanalytics [53,56,57,60,61,62,63,64]. Considering the rapid and sustainable development of biomedical polymers that will directly contact a biological environment, a surface modification method would be one of the most effective ways to introduce specific functionality into these polymers. [52,56,61,65,66,67,68].

In this study, bamboo charcoal (BC) was modified by an O_2_ plasma treatment to form peroxide groups on the surface. Hydrophilic bamboo charcoal was grafted thermo-sensitive antibacterial hydrogel under UV light on the surface. Silver ions (Ag^+^) will be used to improve the antibacterial activity and the surface area of bamboo charcoal. UV-VIS and XRD supported the formation and distribution of AgNPs. The property of the modified bamboo charcoal was characterized by SEM, FTIR, XPS, swelling ratio, and wettability for surface functionalization. Antibacterial activities and biocompatibility of these complexes were investigated. The above results showed a novel idea for a fast and novel method used to prepare thermo-sensitive antibacterial hydrogel. In addition, it could offer reference for the future development and application of the surface modification of bamboo charcoal treatment of biomedical applications.

## 2. Materials and Methods

### 2.1. Pretreatment of Materials

Bamboo charcoal (Taiwan, Nantou, 100% Moso bamboo) samples were cut into a suitable size of about 10 mm × 10 mm × 2 mm. The substrates were ultrasonic-cleaned with ethanol and deionized water for 15 min in each solution before O_2_ plasma treatment to remove surface contamination, and dried within an oven under 40 °C overnight.

### 2.2. Synthesis of Silver Nanoparticles (AgNPs)

AgNPs were first synthesized by photochemical reduction at UV irradiation (model UCE-200s manufactured by Junsun Tech Co., Ltd. New Taipei, Taiwan) having a wavelength of 365 nm over 2 min using a mixing solution of silver nitrate (AgNO_3_, Mw: 169.87 g/mol) to silver ions used trisodium citrate dihydrate (Na_3_C_6_H_5_O_7_·2H_2_O, Mw: 294.10 g/mol) as a reducing agent. Polyvinylpyrrolidone (PVP, (C_6_H_9_NO)_n_, Mw: 40,000 g/mol) is an excellent stabilizer for preventing particle aggregation. The UV source was a mercury lamp operating at 1000 W. After UV irradiation, the solution was centrifuged at 12,000 rpm for 15 min and then filtered to obtain the pellets of AgNPs. Table 1 lists the feed compositions of AgNPs.

### 2.3. O_2_ Plasma Activation Pre-Treatment

In this study, radio frequency (RF) oxygen plasma (model UDS manufactured by Junsun Tech Co., Ltd. New Taipei, Taiwan) was used to process the BC graft onto the substrate. A vacuum pump was used to provide a low-pressure environment. Cleaned BC specimens were placed on the lower electrode of the reaction chamber before being evacuated. The reaction chamber was evacuated to less than 9 mTorr. Bamboo charcoal substrates were subjected to O_2_ plasma pretreatment to form peroxide groups and activate groups on the surface. The processing power was at 25 W and 100 W for a treatment time of 3 min, 20 sccm, respectively. The BC substrate, after oxygen plasma surface activation, has free radicals and hydrophilic functional groups. The significant advantage of plasma surface modification is that activating atoms to ions will not change the substrate properties but improve the substrate’s wettability. At the same time, the free radicals quickly react with organic matter.

### 2.4. Thermo-Sensitive AgNPs Hydrogel by UV Light Surface Graft Polymerization

The O_2_ plasma-treated BC specimen was soaked in an aqueous solution of mixed 10 mmol NIPAAm (H_2_C=CHCONHCH(CH_3_)_2_, Mw = 113.16 g/mol) monomer and S2 solution and vitamin B_2_ with a ratio of monomer solution to B_2_ of 4:1 in a Pyrex glass utensil. The aqueous solution also contained 5 mol% of N′-methylene-bis-acrylamide (NMBA, (H_2_C=CHCONH)_2_CH_2_, Mw = 154.17 g/mole), 1 mol % of Ammonium Persulfate (APS,(NH_4_)_2_S_2_O_8_,Mw=228.20 g/mole), and 1 mol % of (TEMED, (CH_3_)_2_NCH_2_CH_2_N(CH_3_)_2_, Mw = 116.20 g/mole) with NIPAAm monomers. The Pyrex glass utensil was sealed and then irradiated with UV light (wavelength of 365 nm) at 1000 W to induce grafting polymerization from thermo-sensitive AgNPs hydrogels. The irradiation process was done for 1, 2, 3, 4, and 5 min. After graft polymerization, the grafted specimens were washed with distilled water overnight to remove the homopolymer aqueous solution. Figure 1 shows the schematic illustration of the preparation of the functionalization of BC surface modification.

### 2.5. Characterization Analysis

#### 2.5.1. UV-VIS Spectra

The formation of AgNPs and thermo-sensitive AgNPs hydrogels was confirmed using a UV-Visible spectrophotometer (U-2900 Spectrophotometer, HITACHI, Tokyo, Japan) in the wavelength range of 300–700 nm to obtain the UV-Visible spectra of the AgNPs sample, with distilled water as a reference. 

#### 2.5.2. Wettability (Surface Hydrophobicity/Hydrophilicity) Test

The surface wettability of the untreated bamboo charcoal substrates and the O_2_ plasma treatment were measured by the sessile drop (0.9 μL) method with distilled water by a syringe and observed by CCD at room temperature (Dino-Lite AM211, made by AnMo Electronics Corporation, Hsinchu, Taiwan). A video camera recorded the dropped image. The measured water contact angles value was the average of three measurements.

#### 2.5.3. Surface Characterization

The X-ray diffraction measurement (PANalytical X’Pert PRO MPD) is measured to analyze the crystal phase of the AgNPs. Surface morphology of the BC substrates was observed using a scanning electron microscope (JEOL JSM-6701F, Tokyo, Japan). BC substrates were placed on an aluminum holder and sputtering coated into a thin layer of gold (coating 90 s) to improve the electric conductivity. A Fourier transform infrared spectrometer (Jasco FTIR-6200, Tokyo, Japan) was used to analyze the surface functional groups after the surface modification. A ULVAC-PHI PHI 5000 Versaprobe II (Kanagawa, Japan) was used to obtain chemical composition and bonding with modified BC substrate surface through chemical analysis of electron spectroscopy. All the binding energy of photoelectrons at the emission angle was referenced to a CHx peak at the maximum resolved C1s peak at 285.0 eV.

#### 2.5.4. Swelling Studies of the Treatment BC

The weight of dry treatment BC (W_d_) was first measured. Next, the dry treatment BC was placed in solutions of RO water and simulated body fluid (SBF) at 25 °C, 37 °C, and 42 °C to allow the treatment BC to reach an equilibrium swelling state. Following the setting at the temperature of 1 min, 3 min, 5 min, 10 min, 30 min, 60 min, 120 min, 240 min, 480 min, 720 min, 1440 min, 2880 min, and 4320 min, each treatment’s BC weight (Ws) was measured. The swelling ratio (SR) of the treatment BC was recorded during swelling at regular intervals (Equation (1)).
SR = [(W_d_ − W_s)_/W_s_] × 100%(1)
where W_d_ is the weight of the swelling treatment BC at different time points, and W_s_ is the weight of the dry treatment BC.

### 2.6. Cytotoxicity Test of Treatment BC

The cytotoxicity test of surface modification was evaluated by in vitro cell culture, where Alamar Blue was used to measure NIH-3T3 cell viability. Alamar Blue is a cell viability assay reagent that contains a cell-permeable, non-toxic, and weakly fluorescent blue indicator dye called resazurin [64,65]. The samples were washed with phosphate-buffered saline (PBS) solution and placed in a 24-well plate. NIH-3T3 cells were prepared in Dulbecco’s Modified Eagle medium containing 10% fetal bovine serum and 100 U/mL penicillin-streptomycin-amphotericin and inoculated directly onto the sample at a density of 3 × 10^4^ cells/mL They were then keept in a gas-jacketed incubator with 5% CO_2_ at 37 °C. The cell culture time was set to 1, 3, 5, and 7 days at 37 °C for direct cell viability determination. Then, Alamar Blue measurement solution was added to each well, the petri dish wrapped with aluminum foil, and incubated at 37 °C for 4 h. After four hours, an enzyme-linked immunosorbent assay reader was used to measure the optical density (OD) at a wavelength of 570 nm. Cell viability determination is expressed as mean ± standard deviation (*n* = 3). 

### 2.7. Antibacterial Efficacy Test

The bacterial strains used in this study were Escherichia coli (*E. coli*, ATCC, strain 25922) of Bioresource Collection and Research Center (BCRC, Hsinchu, Taiwan). The Kirby-Bauer test tested the antimicrobial effect. The *E. coli* were incubated for 17 h with vigorous shaking (250 rad/min) at 37 °C. The presence or absence of bacteria (turbidity) was determined by the spectrophotometer optical density (OD) measurements at a wavelength of 600 nm after shaking. Each set of samples was inoculated with 0.4 mL (10^7^ CFU/mL) of the bacterial suspension. Afterward, different pairs of treatment BC substrates with a diameter of 10 mm were applied on the surface of the medium. After 24 h and 48 h of incubation at 37 °C, the antibacterial properties against *E. coli* were evaluated by measuring the diameter of the inhibition zone to the sample area. If the bacteria were susceptible to the treatment BC substrates, a zone of inhibition appeared on the agar plate. If it was resistant to the treatment BC substrates, then no zone was evident. All experiments for antibacterial effects have been performed in triplicate: the mean of three replicates was taken for each strain. All measurements were performed in triplicate and averaged. The inhibition diameter of each sample yields significantly statistical meaning with *p* < 0.05 with the control group.

## 3. Results and Discussion

### 3.1. UV-Visible Spectroscopy Characterization of the AgNPs and Thermo-Sensitive AgNPs Hydrogels

According to the surface plasmon resonance (SPR) effect, silver nanoparticles exhibit unique physical properties, depending on the nanoparticles’ shape, size, and distribution [16,19,21,26,27,63]. Therefore, metallic nanoparticles have characteristic optical absorption spectra in the UV–vis region, which varies in the 300−700 nm range [16,19,21,26,27,63]. AgNPs form a broad surface plasma resonance ultraviolet absorption band in the 390–450 nm wavelength range.

The synthesis of AgNPs was carried out by the chemical reduction method. It used trisodium citrate dihydrate as a reducing agent and protective agent, while Polyvinylpyrrolidone (N-vinylpyrrolidone, PVP) is used as a stabilizer. The silver ions (Ag^+^) in silver nitrate (AgNO_3_) solution are reduced to silver atoms in UV irradiation. It is not easy to reunite with other ions through the protective function of trisodium citrate dihydrate. Figure 2a shows the typical UV-vis spectra of colloidal AgNPs with different initial trisodium citrate dihydrate concentrations (10 mM, 50 mM, and 100 mM). In addition, the size distribution of nanoparticles with different concentrations of trisodium citrate dihydrate, the colloidal of AgNPs was prepared by 10 mM gave 65 nm, 50 mM gave 75 nm, and 100 mM gave 89 nm. It was confirmed that the size of NPs was also reduced as the trisodium citrate dihydrate concentration decreased from 100mM to 10mM. Therefore, the S2 solution that gave the smallest size was selected as the optimized ratio. In the study, AgNPs were synthesized by UV irradiation successfully. The golden color show confirmed the formation of AgNPs in Figure 2a. The existence of this color change can be observed stability and size of AgNPs.

Figure 2b illustrates the UV-Vis absorption spectra of the thermo-sensitive AgNPs hydrogel grafted with different UV irradiation times. The thermo-sensitive AgNPs hydrogel that exhibited a sharp, narrow, intense peak of maximum absorption had a UV irradiation time of 5 min, as shown in Figure 2b. According to the UV-Vis absorption spectrum, it can be inferred that the longer the time for graft polymerization of the thermo-sensitive AgNPs hydrogel, the position of the absorption peak of the hydrogel solution will also change. The reason may be the change of the colloidal structure, which affects the metal’s free electrons in the hydrocolloid solution resonating with the reflected light to produce oscillations. Therefore, the UV irradiation time must be controlled to produce stable colloidal of thermo-sensitive AgNPs hydrogel. The thermo-sensitive AgNPs hydrogel irradiated with UV light for 5 min is the best hydrogel obtained after optimizing the reaction conditions, so it is selected to study all subsequent reaction parameters. The indication that the thermo-sensitive AgNPs hydrogel aggregates to a larger size is the color of the solution changing from yellow to dark yellow. The stability of thermo-sensitive AgNPs hydrogel is observed by the color change shown in Figure 2c. The light transmittance can express the phase difference of the thermo-sensitive AgNPs hydrogel at the LCST temperature. As shown in Figure 2d, the color of the thermo-sensitive AgNPs hydrogel begins to change around 32 °C. As the temperature increased (>32 °C), the thermo-sensitive AgNPs hydrogel gradually changed from a yellow transparent state to an opaque white state, and the light transmittance decreased significantly.

### 3.2. XRD Characterization of Thermo-Sensitive AgNPs Hydrogels

Figure 3 shows the XRD patterns of the thermo-sensitive AgNPs hydrogels at a UV irradiation time of 5 min.

The results showed that the crystalline nature of the thermo-sensitive AgNPs hydrogel was confirmed to be crystalline silver. The four diffraction peaks of the thermo-sensitive AgNPs hydrogel at 38.10°, 43.47°, 65.39°, 78.23°, and 81.5° are related to the face-centered cube structure of AgNPs in the (111), (200), (220), and (311) planes [16,19,21,26,27,63], respectively, confirmed the presence of AgNPs in the thermo-sensitive hydrogel nanocomposite. 

### 3.3. Wettability of Surface-Modified Bamboo Charcoal

The surface wettability of the modified substrate can be measured by the water contact angles (WCA). The measured water contact angle of un-modified BC is 63.5 ± 7.8°. The values of the water contact angle of BC substrate after each treatment are listed in Table 2. The results were observed that the BC surface became extremely hydrophilic after O_2_ plasma treatment (BC-O_2_-25W and BC-O_2_-100W). The treatment of O_2_ plasma on BC substrate introduces polar functional groups such as hydroxyl and carboxyl. The BC surface still maintains hydrophilic water after thermo-sensitive AgNPs hydrogels grafting (BC-O_2_-25W-g and BC-O_2_-100W-g).

### 3.4. Swelling Ratio of Surface-Modified Bamboo Charcoal

Table 3 showed the SR (%) variations of BC substrate grafted with the thermo-sensitive AgNPs hydrogels. In the swelling ratio, both control and thermo-sensitive AgNPs-grafted hydrogels reached equilibrium after about 72 h. The data in the table show that the thermo-sensitive AgNPs-grafted hydrogels exhibit thermo-sensitive properties. Regardless of the solution of RO water or SBF, the swelling ratio decreases when the temperature increases from 28 °C to 37 °C. The swelling of the hydrogel may be closely related to the mesh size and consequently to the permeability of the hydrogels.

### 3.5. FTIR Characterization of Surface-Modified Bamboo Charcoal 

FTIR measurements were carried out to study the differences in the modified chemical structure of BC substrate with different treatments. As Figure 4 shows of (a) un-modified, (b) O_2_ plasma treatment (100 W), and (c) O_2_ plasma treatment (100 W) +UV graft thermo-sensitive AgNPs hydrogels BC specimens. It could be observed that several adsorption peaks after O_2_ plasma treatment appeared, such as, for example, -OH at 3050~3250 cm^−1^, C=O at 1700–1720 cm^−1^, and C-O at 1140 cm^−1^, were reduced [37,38,39,40,41]. The increase in these oxygen-containing functional groups revealed the due to oxygen atom bonding to hydrogen atoms on the surface during O_2_ plasma treatment [37,38]. The functional groups exposed by O_2_ plasma treatment made the hydrophilicity and surface energy could be increased [37,38,39]. After the UV-grafted with thermo-sensitive AgNPs hydrogels, the functional group N-H peak of NIPAAm was found at 3200~3600 cm^−1^, O-H peak was found at 3000 cm^−1^, C–H was found at 1340–1430 cm^−1^, and C=O peak was found at 1700 cm^−1^ [54,56,57,58,59]. All the above-observed peaks in the spectrum revealed the significant groups associated with NIPAAM chemical structures, indicating the successful attachment of thermo-sensitive hydrogels onto the BC substrate. In addition, no new functional groups were added or disappeared in the FT-IR spectrum. It can be proved the introduction of AgNPs did not affect the structure of PNIPAAm. Thermo-sensitive AgNPs hydrogels were only physically connected without chemical bonding.

### 3.6. Chemical Composition Analysis of Surface-Modified Bamboo Charcoal

Chemical and elemental surface characterization by X-ray photoelectron spectroscopy (XPS) is a convenient and sensitive method of investigating surface modification, as shown in Figure 5. This study used an O_2_ plasma treatment (100 W) as the substrate for ESCA analysis. The binding energy (BE) and peak intensity of the spectra were calibrated and normalized by C1s, O1s, N1s, Ag3d originating from the BC substrates. The surface chemical composition of control and treated BC was evaluated. The XPS analyses of the C1s peak were deconvolution into three peaks corresponding to C-C (284.4 eV), C-O (285.4 eV), and O=C-O (288.6 eV). The O1s peak was deconvolution into three peaks corresponding to O-C (532.7 eV), O=C (531.6 eV), and O-C=O (530.0 eV). After the O_2_ plasma treatment, the peaks of C1s and O1s, which are the constituent elements of oxygen, were detected on the specimen surface. The C1s XPS spectrum of the BC surface can be fitted by three peaks, each representing a separate C bond; C-C/C-H (284.4 eV), C-O (285.4 eV), and O=C-O (288.7 eV). The O1s XPS spectrum of the BC surface can be fitted by three peaks, each representing a separate O bond; O=C/O=C-O (530.0 eV), O-C (531.6 eV), and C-OH (532.3 eV). The O_2_ plasma specimens show an increase in carbon-oxygen functionalities relative to the as-received BC specimens. The XPS analysis also measured the chemical compositions of the grafted thermo-sensitive AgNPs hydrogels; Figure 5 shows that the C1s XPS spectrum of thermo-sensitive AgNPs hydrogels grafted onto the surface of O_2_ plasma-treated BC substrate can be fitted by three peaks, C-C/C-H (283.7 eV), C-N (284.8 eV), and -C=O (286.5 eV), respectively, the O1s can be fitted by three peaks, O=C-N (529.8 eV), -C=O (530.0 eV)and C-O-C (530.7eV), the N1s can be the fitted by three peaks, -NH (398.2 eV), O=C-N (398.5 eV) and C-N (399.1 eV), and the Ag3d can be the fitted by two peaks, Ag3d_5/2_ (367.2 eV) and Ag3d_3/2_ (373.5 eV). Peak component C-N was associated with the carbon atom attached directly to the nitrogen atom in the amino group. It was attested that thermo-sensitive AgNPs hydrogels grafted onto the BC substrate successfully.

### 3.7. Surface Morphology of Surface-Modified Bamboo Charcoal

Figure 6 shows the surface morphology of (a) un-modified, (b) O_2_ plasma treatment (100 W), and (c) O_2_ plasma treatment (100 W) +UV graft thermo-sensitive AgNPs hydrogels BC specimens. The surface of un-modified BC has a porous structure. After being treated by O_2_ plasma, the surface was modified without changing the porous structure. Form Figure 6c shows the surface network SEM micrographs of BC specimens subjected to O_2_ plasma treatment and grafting with thermo-sensitive AgNPs hydrogels. The hydrogel covers the BC surface. The fine and disaggregated silver particles were homogeneously distributed on the surface of the hydrogel. The results can be explained by thermo-sensitive AgNPs hydrogels formed on the BC surface.

### 3.8. In Vitro Cytocompatibility Assay of Surface-Modified Bamboo Charcoal

The in vitro cytocompatibility of BC specimens with different surface treatment conditions was analyzed using the Alamar Blue assay, as shown in Figure 7. To show the unilateral effect of O_2_ plasma treatment (100 W), O_2_ plasma treatment (100 W), and UV graft thermo-sensitive AgNPs hydrogels on cell behavior, nontreated BC specimens were used as controls. On day 3, it was observed that the OD value of O_2_ plasma treatment (100 W) and UV graft thermo-sensitive AgNPs hydrogels increased obviously with the culture time, and the tendencies were even more obvious after 7 days of incubation. There was no apparent cytotoxicity after O_2_ plasma treatment and UV graft thermo-sensitive AgNPs hydrogels. In this study, the cells spread well and displayed a standard shape in the coatings incorporating silver, indicating that these implanted doses of Ag^+^ are nontoxic to the colony cells. In that sense, thermo-sensitive AgNPs hydrogels on the BC specimens are sufficient to destruct bacterial cells without inducing notable cytotoxicity.

### 3.9. Antibacterial Characteristics of Surface-Modified Bamboo Charcoal

The antibacterial efficacy of the surface-modified BC was tested based on the zone of inhibition. Figure 8 and Table 4 show the bacterial colonies as clear transparent rings after 24 h, with a zone of inhibition obtained around O_2_ plasma treatment (100 W) and UV graft thermo-sensitive AgNPs hydrogels showing the inhibiting effect on bacterial activity. After 24 h of incubation, the zones of inhibition of the thermo-sensitive AgNPs hydrogels against *E. coli* ranged 15.7 ± 0.2 mm, whereas BC did not show any zone of inhibition. This value (15.7 ± 0.2 mm) represents the diameter of the zone of inhibition. It is representing the average diameter of the inhibition zone. According to Standard ‘SNV 195920-1992′, the procedure consisted of incubating the substrate for 24 h at 37 °C in contact with *E. coli* on nutrient agar plates and then evaluating the presence of an area of inhibited bacterial growth around the substrate. The antibacterial capability of the substrate was defined as a function of the width of the inhibition area, according to the levels provided by the Standard. Thus, if the width of the bacterial inhibition area is greater than 1 mm, a “good” antibacterial activity can be associated with the substrate; on the other hand, if bacteria fully cover the sample, its antibacterial activity is labeled as “insufficient” [69,70]. The O_2_ plasma treatment (100 W) and UV graft thermo-sensitive AgNPs hydrogels on BC samples showed a clear inhibition zone, which indicated the efficient antibacterial ability of these samples. In this study, the released silver ions were responsible for bactericidal activity of the BC surface by the destructing bacterial cell membrane. Samples fabricated via surface treatment exhibited a good antibacterial effect against bacteria. This study demonstrates their potential use as antibacterial coating materials for biomedical applications.

## 4. Conclusions

The results show AgNPs and thermo-sensitive AgNPs hydrogels can be synthesized easily and at a meager cost by UV irradiation. A thermo-sensitive AgNPs hydrogel was prepared by combining PNIPAAm with AgNPs solution. The yield is clean and without chemical contamination, ensuring it is suitable for biomedical applications. Optical measurements of AgNPs and thermo-sensitive AgNPs hydrogels optical ranged from 418 nm to 429 mm and from 414 to 427 mm, respectively, related to surface plasmon resonance. This study showed that in the O_2_ plasma treatment with BC, the wettability results show that the substrate following the O_2_ plasma treatment improves the hydrophilic quality of substrate efficiency. Then, thermo-sensitive AgNPs hydrogels were combined by UV light surface graft polymerization on BC. The FTIR spectra and ESCA composition analysis prove that thermo-sensitive AgNPs hydrogels could be immobilized on the specimen BC surface. The swelling ratios of the thermo-sensitive AgNPs hydrogels at a lower temperature are larger than those at a higher temperature environment. Additionally, the thermo-sensitive AgNPs hydrogels integrated well with the BC without damaging the BC structure.

In summary, the results suggested that O_2_ plasma treatment (100 W) and UV graft thermo-sensitive AgNPs hydrogels on BC exhibit good antibacterial properties against bacteria. Due to the presence of silver ions, the modified BC had a good antibacterial effect. The BC was modified by a plasma treatment and UV-graft hydrogel to have both temperature-sensitive and antibacterial properties, promising biomedical applications.

## Figures and Tables

**Figure 1 nanomaterials-11-02697-f001:**
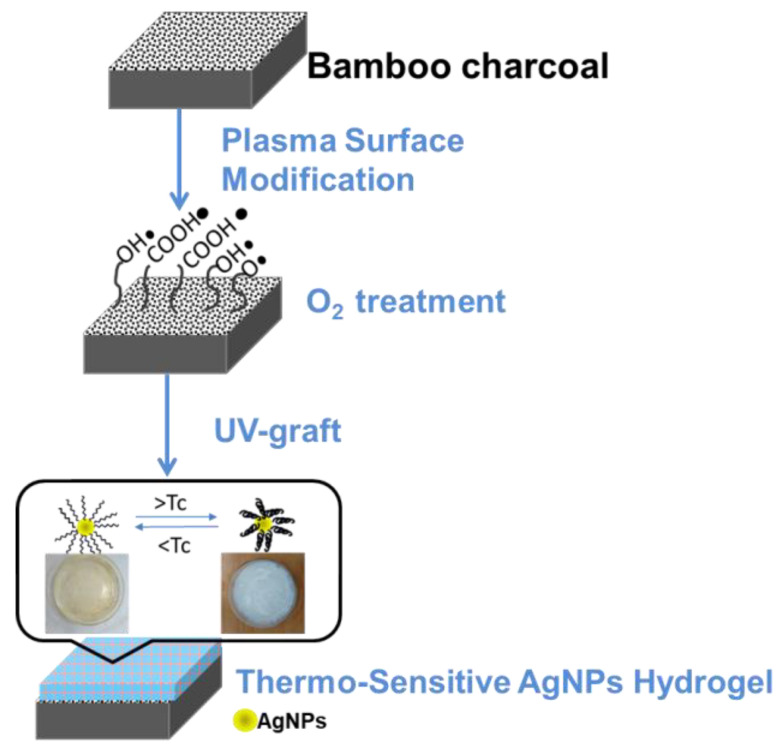
The schematic illustration of the preparation of the functionalization of surface modification with O_2_ plasma pretreatment and then graft polymerization of a Thermo-sensitive AgNPs hydrogel onto the BC.

**Figure 2 nanomaterials-11-02697-f002:**
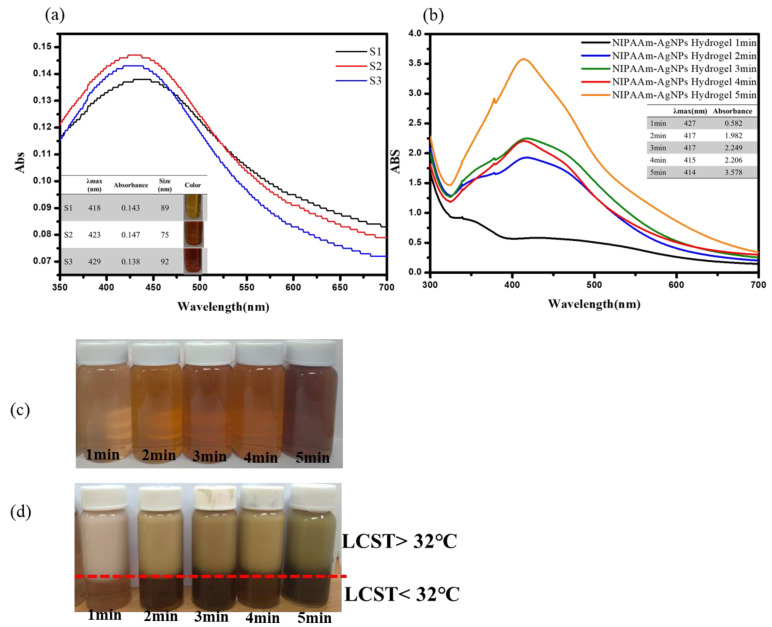
(**a**) UV-vis spectra maxima of the AgNPs. (**b**) UV-vis spectra maxima of the thermo-sensitive AgNPs hydrogels at a different time. (**c**) The color change of thermo-sensitive AgNPs hydrogels. (**d**) Photographs of the LCST effect of thermo-sensitive AgNPs hydrogels.

**Figure 3 nanomaterials-11-02697-f003:**
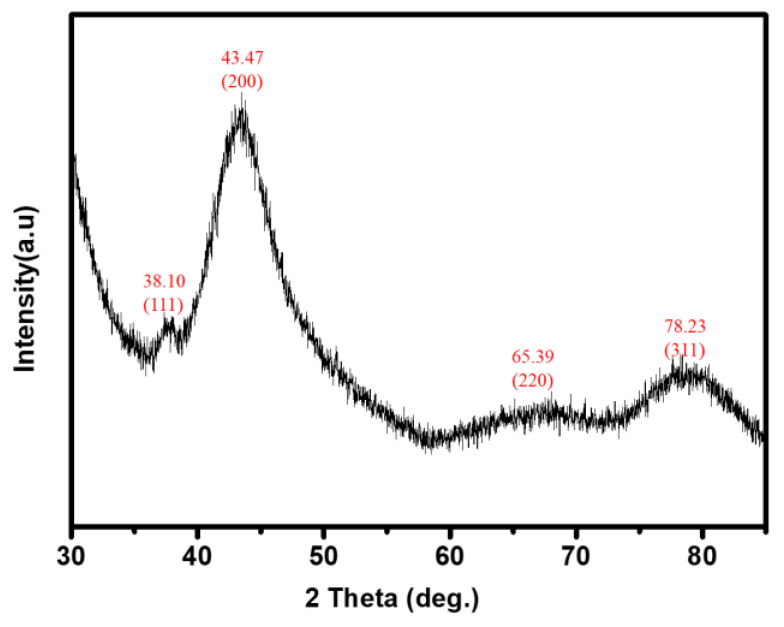
XRD spectra for thermo-sensitive AgNPs hydrogels obtained after 5 min of UV irradiation.

**Figure 4 nanomaterials-11-02697-f004:**
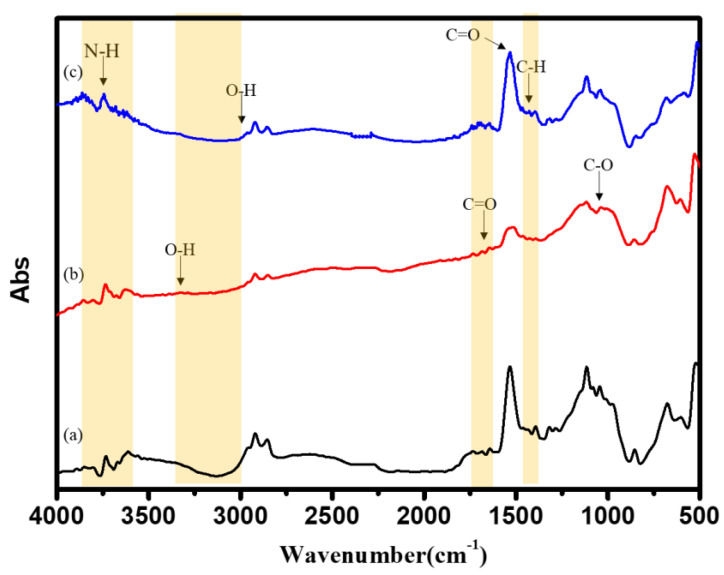
Fourier Transformation Infrared (FTIR) spectra of (**a**) un-modified, (**b**) O_2_ plasma treatment (100 W), and (**c**) O_2_ plasma treatment (100 W) +UV graft thermo-sensitive AgNPs hydrogels BC specimens.

**Figure 5 nanomaterials-11-02697-f005:**
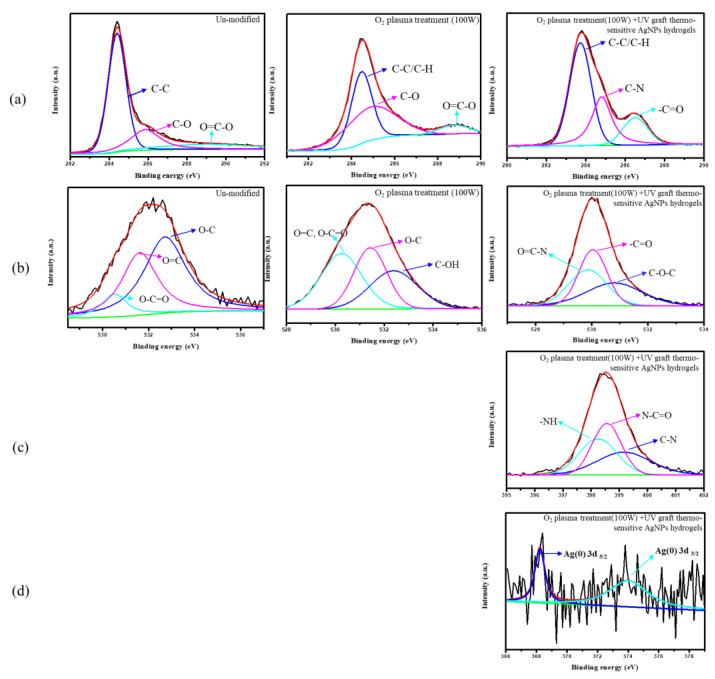
The XPS spectra of BC specimens for different treatment (**a**) C 1s spectra, (**b**) O1s spectra, (**c**) N 1s spectra (**d**) Ag3d spectra.

**Figure 6 nanomaterials-11-02697-f006:**
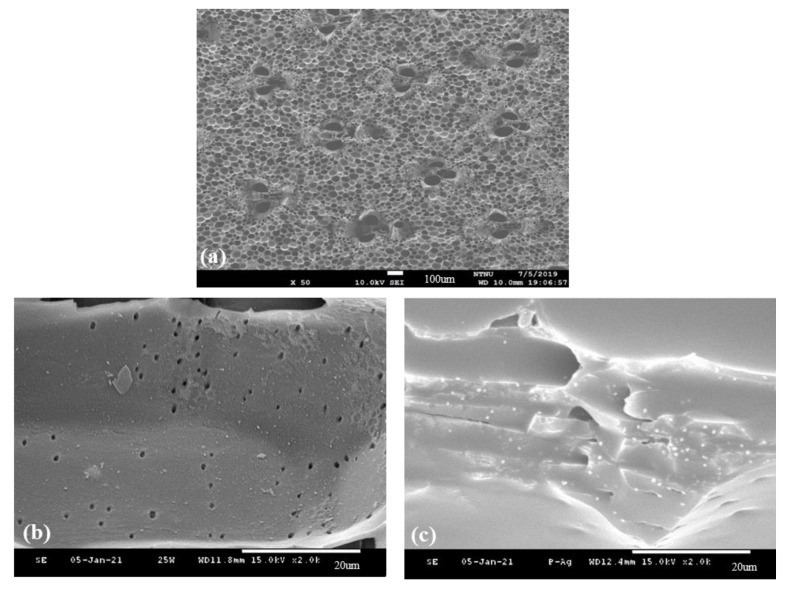
Surface morphologies of (**a**) un-modified, (**b**) O_2_ plasma treatment (100 W), and (**c**) O_2_ plasma treatment (100 W) +UV graft thermo-sensitive AgNPs hydrogels BC specimens.

**Figure 7 nanomaterials-11-02697-f007:**
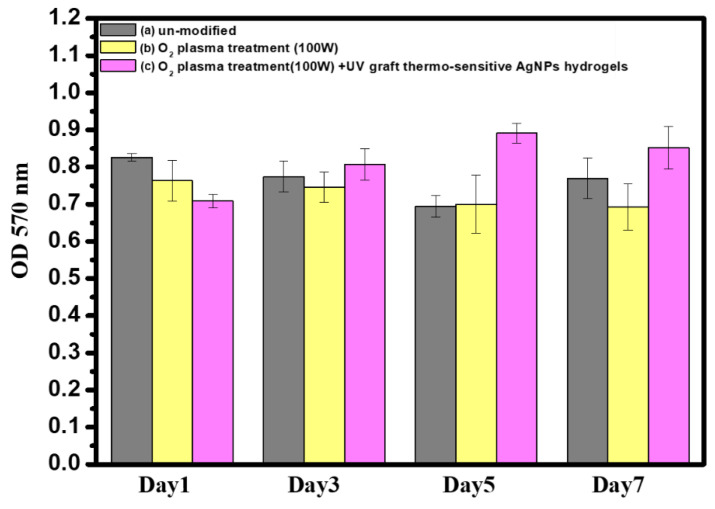
Cytocompatibility Assay of (**a**) un-modified, (**b**) O_2_ plasma treatment (100 W), and (**c**) O_2_ plasma treatment (100 W) +UV graft thermo-sensitive AgNPs hydrogels BC specimens over a period of 1 to 7 days. (Error bars mean ± standard deviation (*n* = 3)).

**Figure 8 nanomaterials-11-02697-f008:**
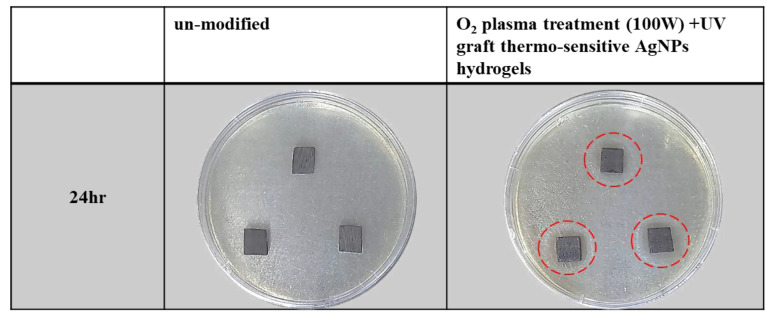
Photographs of the antibacterial test results on *E. coli*.

**Table 1 nanomaterials-11-02697-t001:** Feed composition of AgNPs.

Type	AgNO_3_ (mM)	Na_3_C_6_H_5_O_7_ (mM)	PVP (μM)
S1	10	10	75
S2	10	50	75
S3	10	100	75

**Table 2 nanomaterials-11-02697-t002:** Water contact angles of BC substrate after different treatment.

	Untreated	Treatment A	Treatment B	Treatment C	Treatment D
θ_H__2__O_	63.5 ± 7.8°	<0°	<0°	32.1 ± 1.1°	34 ± 1.5°

Treatment A-O_2_ plasma treatment (25 W), treatment B-O_2_ plasma treatment (100 W), treatment C-O_2_ plasma treatment (25 W) +UV graft thermo-sensitive AgNPs hydrogels, treatment D-O_2_ plasma treatment (100 W) +UV graft thermo-sensitive AgNPs hydrogels.

**Table 3 nanomaterials-11-02697-t003:** Variations of SR (%) for different solution tested from 28 °C to 37 °C.

		**Temperature (°C)**		
		25	37	42
RO water	Untreated	103.2	103.9	106.9
	A	91.5	69.8	72.6
	B	134.8	51.2	54.8
		**Temperature (°C)**		
		25	37	42
SBF solution	Untreated	39.6	33.7	33.0
	A	64.7	59.8	54.3
	B	87.8	71.1	67.8

Treatment A-O_2_ plasma treatment (25 W) +UV graft thermo-sensitive AgNPs hydrogels, treatment B-O_2_ plasma treatment (100 W) +UV graft thermo-sensitive AgNPs hydrogels.

**Table 4 nanomaterials-11-02697-t004:** Zone of inhibition in agar diffusion tested against the surface-modified BC after 24 h.

Test Organism	Diameter Zone (mm), Mean (*n* = 3)	
	Un-Modified	O_2_ Plasma Treatment (100 W) +UV Graft Thermo-Sensitive AgNPs Hydrogels
*E. coli*	0	15.7 ± 0.2 (mm)

## Data Availability

The study did not report any data.

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
