# Peer review of "Surface Modification of Bamboo Charcoal by O2 Plasma Treatment and UV-Grafted Thermo-Sensitive AgNPs Hydrogel to Improve Antibacterial Properties in Biomedical Application"

_nanomaterials, 2021, doi:10.3390/nano11102697_

Round 1
Reviewer 1 Report
Dear authors:
The work in your manuscript is interesting and the writing is generally clear with some minor problems separating numbers and units or grammatical signs that can be corrected during professional edition.
In my opinion you need to more clearly distinguish in the text and Figure 8 the differences between the tested sample and its control. In addition, it is customary in this kind of research to report the minimum inhibitory concentration, which is missing from your work. Table 4 is missing the units for 15.7 mm. Explain better in the text what is the meaning of this value and discuss if it is significantly different at a 95% confidence level from its control. I am having a hard time to appreciate the value of this test as there is no perceptible color changes.
Author Response
Dear Editors and Reviewers,
Thank you for your letter and for the reviewers’ comments concerning our manuscript entitled “Surface Modification of Bamboo Charcoal by O2 Plasma Treatment and UV-grafted Thermo-Sensitive AgNPs Hydrogel to Improve Antibacterial Properties in Biomedical Application” (nanomaterials-1386351). Those comments are all valuable and very helpful for revising and improving our paper and the important guiding significance to our research. We have studied comments carefully and have made a correction that we hope to meet with approval. A revised manuscript with the correction sections yellow marked was attached as the supplemental material and for easy check/editing purposes. Should you have any questions, please get in touch with us without hesitation. It is hoped that after the revision, the manuscript meets the standard for publication in your journal. The major corrections in the paper and the response to the reviewer’s comments are as follows:
Response to Reviewers
Reviewer
The work in your manuscript is interesting and the writing is generally clear with some minor problems separating numbers and units or grammatical signs that can be corrected during professional edition.
In my opinion you need to more clearly distinguish in the text and Figure 8 the differences between the tested sample and its control. In addition, it is customary in this kind of research to report the minimum inhibitory concentration, which is missing from your work. Table 4 is missing the units for 15.7 mm. Explain better in the text what is the meaning of this value and discuss if it is significantly different at a 95% confidence level from its control. I am having a hard time to appreciate the value of this test as there is no perceptible color changes.
Reply:
Thank you for your mention. I am very grateful for your comments about the manuscript. According to your advice, we amended the relevant part of the manuscript. Some of your questions were answered below.
(i) As reviewer suggested that we already added this information in the manuscript.
(ii) Regarding the English language, we revised the document and improved it.
- In my opinion you need to more clearly distinguish in the text and Figure 8 the differences between the tested sample and its control.
Reply:
Thanks for your comment. About the inhibition zone test in this study, the untreated bamboo charcoal (BC) test piece in figure 8 does not have any antibacterial material on the surface, so no clear zone (the zone of inhibition) was observed on the agar plate. But the BC after surface modification, AgNPs leaches from the BC into the agar and then exert a growth-inhibiting effect. A clear zone (the zone of inhibition) appears around the test product on the agar plate on the right-hand side of the photo. The size of the zone of inhibition is usually related to the level of antimicrobial activity present in the sample or product - a larger zone of inhibition usually means that the antimicrobial is more potent. We adjusted the brightness of the antibacterial result photos to make them easier for readers to see.
- In addition, it is customary in this kind of research to report the minimum inhibitory concentration, which is missing from your work.
Reply:
Thank you for your mention. We have added minimum inhibitory concentration in the manuscript. Thermo-sensitive AgNPs hydrogels have 90ppm AgNPs had minimum inhibitory concentration.
- Table 4 is missing the units for 15.7 mm.
Reply:
Thank you for your mention. The unit has been added in Table 4.
- Explain better in the text what is the meaning of this value and discuss if it is significantly different at a 95% confidence level from its control. I am having a hard time to appreciate the value of this test as there is no perceptible color changes.
Reply:
Thanks for your comment. This value (15.7 ±0.2mm) represents the diameter of the zone of inhibition. It represents the average diameter of the inhibition zone. All experiments for antibacterial have been performed in triplicate: the mean with three replicates taken for each strain. All measurements were performed in triplicate and averaged. The agar diffusion tests were conducted according to Standard ‘SNV 195920-1992’. The procedure consisted in incubating the samples for 24 h at 37 Ì‹C in contact with bacteria on nutrient agar plates and then evaluating the presence of an area of inhibited bacteria growth around the samples. The antibacterial capability of the sample was defined as a function of the width of the inhibition area, according to the levels provided by the Standard. Thus, if the width of the bacterial inhibition area is greater than 1 mm, a “good” antibacterial activity can be associated with the sample; on the other hand, if bacteria fully cover the sample, its antibacterial activity is labeled as “insufficient” [R1-R2]. These texts have been added to the section 2.7 and 3.9.
[R1] Paladini, F., Di Franco, C., Panico, A., Scamarcio, G., Sannino, A. and Pollini, M., 2016. In Vitro Assessment of the Antibacterial Potential of Silver Nano-Coatings on Cotton Gauzes for Prevention of Wound Infections. Materials, 9(6), p.411.
[R2] Pollini, M., Russo, M., Licciulli, A., Sannino, A. and Maffezzoli, A., 2009. Characterization of antibacterial silver coated yarns. Journal of Materials Science: Materials in Medicine, 20(11), pp.2361-2366.
We tried our best to improve the manuscript and made some changes in the manuscript. These changes will not influence the content and framework of the paper. And here we did not list the changes but marked in yellow in revised paper.
We appreciate for editors/reviewers’ warm work earnestly, and hope that the correction will meet with approval.
Once again, thank you very much for your comments and suggestions.
Sincerely yours,
Shu-Chuan Liao

Reviewer 2 Report
In this manuscript, the author reports, ‘Surface Modification of Bamboo Charcoal by O2 Plasma Treatment and UV-grafted Thermo-Sensitive AgNPs Hydrogel to Improve Antibacterial Properties in Biomedical Application’. The current study is on a topic of relevance and general interest to readers in this area. The authors should address the following questions before getting a possible publication.
Recommendation: Major revisions needed as noted.
- In Figure 6, the scale bars are not clearly visible to the readers.
- The author should write the purpose for each test in one/two sentences (in brief) before explaining the results of the characterization techniques. Therefore, the logic and organization of this part will be enhanced
- The novelty of the present work should be discussed in the Introduction section.
- What does the error bars stand for presented in the inset of Figure 7? It should be mentioned in Figure captions.
- In section 3.9., the authors stated that the zone of inhibition of the thermo-sensitive AgNPs hydrogels against E. coli ranged is 7±0.2 mm. What does this value stands for? diameter or radius?
- The formatting and grammatical errors in the article need to be checked carefully.
- The conclusion should be precise and compact.
- The authors have cited relevant references in the Introduction section; however there are few that need to be included: Molecular pharmaceutics, 6(5), 1388-1401; Ultrasonics sonochemistry 60 (2020): 104797; Materials Science and Engineering: C 92 (2018): 575-589; ACS Applied Materials & Interfaces, 12(46), 51940-51951
Author Response
Dear Editors and Reviewers,
Thank you for your letter and for the reviewers’ comments concerning our manuscript entitled “Surface Modification of Bamboo Charcoal by O2 Plasma Treatment and UV-grafted Thermo-Sensitive AgNPs Hydrogel to Improve Antibacterial Properties in Biomedical Application” (nanomaterials-1386351). Those comments are all valuable and very helpful for revising and improving our paper and the important guiding significance to our research. We have studied comments carefully and have made a correction that we hope to meet with approval. A revised manuscript with the correction sections yellow marked was attached as the supplemental material and for easy check/editing purposes. Should you have any questions, please get in touch with us without hesitation. It is hoped that after the revision, the manuscript meets the standard for publication in your journal. The major corrections in the paper and the response to the reviewer’s comments are as follows:
Response to Reviewers
Reviewer
In this manuscript, the author reports, ‘Surface Modification of Bamboo Charcoal by O2 Plasma Treatment and UV-grafted Thermo-Sensitive AgNPs Hydrogel to Improve Antibacterial Properties in Biomedical Application’. The current study is on a topic of relevance and general interest to readers in this area. The authors should address the following questions before getting a possible publication.
Reply:
Thank you for your mention. I am very grateful for your comments about the manuscript. According to your advice, we amended the relevant part of the manuscript. Some of your questions were answered below.
(i) As reviewer suggested that we already added this information in the manuscript.
(ii) Regarding the English language, we revised the document and improved it.
- In Figure 6, the scale bars are not clearly visible to the readers.
Reply:
We are very sorry for our negligence in our manuscript. We have updated Figure 6 to visible the scale bars.
2.The author should write the purpose for each test in one/two sentences (in brief) before explaining the results of the characterization techniques. Therefore, the logic and organization of this part will be enhanced.
Reply:
Thank you for your constructive common. We have added one/two sentences before explaining the results of the characterization techniques.
3.The novelty of the present work should be discussed in the Introduction section.
Reply:
Thank you for your constructive common. This study offers a novel idea for a fast and novel method used to prepare silver ion composite hydrogels and silver nanocomposite hydrogels. In addition, it could offer new valuable new opportunities in the surface modification of bamboo charcoal treatment of biomedical application. These texts have been added to the introduction section.
4.What does the error bars stand for presented in the inset of Figure 7? It should be mentioned in Figure captions.
Reply:
Thanks for your comment. The error bars represent the measured OD value was an average of 30 times measurements. (n=3 for each group). It can be seen that, over the 7 days, the cells showed a time-dependent growth pattern on the samples. A statistically significant difference (p < 0.05) in proliferation was seen compared with the untreated BC surface.
5.In section 3.9., the authors stated that the zone of inhibition of the thermo-sensitive AgNPs hydrogels against E. coli ranged is 15.7±0.2 mm. What does this value stands for? diameter or radius?
Reply:
This value (15.7 ±0.2mm) represents the diameter of the zone of inhibition. It represents the average diameter of the inhibition zone. All experiments for antibacterial have been performed in triplicate: the mean with three replicates taken for each strain. All measurements were performed in triplicate and averaged. Zone of inhibition testing is a fast, qualitative means to measure the ability of an antimicrobial agent to inhibit the growth of microorganisms.
6.The formatting and grammatical errors in the article need to be checked carefully.
Reply:
Thank you very much for your suggestion; your suggestion is very pertinent; our English language does need to be improved. By re-reading the manuscript carefully, we have found a lot of phrases and grammatical errors. All of the errors have been revised. We also have invited a native English speaker to proofread the whole paper, and I hope it is more precise and accurate now of this revised paper on the English expression.
7.The conclusion should be precise and compact.
Reply:
Yes, the referee is correct.
The conclusion was more precise and compact discussed.
8.The authors have cited relevant references in the Introduction section; however there are few that need to be included: Molecular pharmaceutics, 6(5), 1388-1401; Ultrasonics sonochemistry 60 (2020): 104797; Materials Science and Engineering: C 92 (2018): 575-589; ACS Applied Materials & Interfaces, 12(46), 51940-51951.
Reply:
Thank you for your mention. We have added the reviewer to provide suitable references in the literature. These texts have been added to the introduction section.
[21] Sood, R. and Chopra, D., 2018. Optimization of reaction conditions to fabricate Ocimum sanctum synthesized silver nanoparticles and its application to nano-gel systems for burn wounds. Materials Science and Engineering: C, 92, pp.575-589.
[28] Jain, J., Arora, S., Rajwade, J., Omray, P., Khandelwal, S. and Paknikar, K., 2009. Silver Nanoparticles in Therapeutics: Development of an Antimicrobial Gel Formulation for Topical Use. Molecular Pharmaceutics, 6(5), pp.1388-1401.
[34] Ganguly, S., Das, P., Das, T., Ghosh, S., Das, S., Bose, M., Mondal, M., Das, A. and Das, N., 2020. Acoustic cavitation assisted destratified clay tactoid reinforced in situ elastomer mimetic semi-IPN hydrogel for catalytic and bactericidal application. Ultrasonics Sonochemistry, 60, p.104797.
We tried our best to improve the manuscript and made some changes in the manuscript. These changes will not influence the content and framework of the paper. And here we did not list the changes but marked in yellow in revised paper.
We appreciate for editors/reviewers’ warm work earnestly, and hope that the correction will meet with approval.
Once again, thank you very much for your comments and suggestions.
Sincerely yours,
Shu-Chuan Liao

Round 2
Reviewer 1 Report
Publish the revised version as is.
Reviewer 2 Report
The authors have addressed all the questions raised before. Therefore the manuscript can be accepted in the present form